# Management of Sporadic Vestibular Schwannomas in Children—Volumetric Analysis and Clinical Outcome Assessment

**DOI:** 10.3390/children9040490

**Published:** 2022-04-01

**Authors:** Julian Zipfel, Mykola Gorbachuk, Isabel Gugel, Marcos Tatagiba, Martin U. Schuhmann

**Affiliations:** Section of Pediatric Neurosurgery, Department of Neurosurgery, University Hospital Tübingen, 72076 Tübingen, Germany; mykola.gorbachuk@med.uni-tuebingen.de (M.G.); isabel.gugel@med.uni-tuebingen.de (I.G.); marcos.tatagiba@med.uni-tuebingen.de (M.T.); martin.schuhmann@med.uni-tuebingen.de (M.U.S.)

**Keywords:** vestibular schwannoma, retrosigmoid approach, volumetry, cerebellopontine angle

## Abstract

Vestibular schwannomas (VS) usually manifest between the 5th and 8th decade of life. Most pediatric cases are associated with Neurofibromatosis type 2 and sporadic VS are rare in this age group. Few case series have been published. We report on our institutional series of sporadic VS in children. We included all cases between 2003 and 2021; 28 of 1635 patients harbored a sporadic VS and were younger than 21 years old. A retrospective review of clinical parameters and surgical data as well as outcomes was performed. All procedures were performed via a retrosigmoid approach. Preoperative imaging was assessed, and tumor volumetry was performed. Mean follow-up was 28 months, symptomatology was diverse. Most children and adolescents presented with hearing loss and tinnitus. All cases with multiple preoperative magnetic resonance imaging scans showed volumetric tumor growth between 1 and 18%/month (mean 8.9 ± 5.6%). Cystic tumor morphology and bone erosion was seen in larger tumors. Gross total resection was possible in 78% of patients and no recurrence was observed. All patients with subtotal resection showed tumor regrowth. Sporadic VS in children are rare and present with a high clinical variability. Surgical resection is the primary therapy and is feasible with favorable results comparable to the adult age group.

## 1. Introduction

Vestibular schwannomas (VS) are benign tumors of the vestibular part of the 8th cranial nerve. In adults, these lesions constitute about 6–8% of all intracranial tumors and 80% of tumors of the cerebellopontine angle (CPA). VS typically manifest clinically via cranial nerve deficits, usually affecting vestibular function and hearing. Larger tumors can lead to facial paresis [1]. In children, approximately 0.8% of tumors are VS [2]. They constitute about 60% of lesions confined to the CPA [3]. Postoperative facial nerve function correlates with tumor volume and auditory function. Hearing can often be preserved in smaller tumors after surgery, however, not in large lesions [4].

Bilateral VS are pathognomonic for Neurofibromatosis type 2 (NF2), a genetic syndrome associated with other peripheral schwannomas, meningiomas and spinal ependymomas. [1] Sporadic VS usually manifest between the 5th and 8th decade of life and are rare in children. Most VS in children are associated with NF2. Sometimes, diagnosis of NF2 is initially missed due to delayed contralateral tumor growth [5]. The most frequent symptoms of NF2 which present first in children are ophthalmological and cutaneous features with VS-associated deficits as a sign of disease severity [6,7].

Most literature on sporadic VS in children are case reports or small case series. There is a small number of reports on very large and highly vascularized tumors [8,9,10,11]. Usually, between 1 and 10 cases are reported in large centers over periods of more than 20 years [8,12].

Epidemiologically, larger sporadic VS are seen more frequently in male patients, especially in the younger population of up to 40 years of age. Furthermore, male patients usually show a higher rate of preoperative hearing loss [13].

Typically, age at diagnosis in children and adolescents is approximately 14 years [14]. Generally, tumors in this age group tend to be larger than in adults at the time of diagnosis [15]. The delay between symptoms and first cranial imaging is shorter than in adults, whereas more NF-mutations have been reported [16]. Non-NF2-VS also show aberrations of the NF2-gene without germline affection [17].

A case series from Bergamo reported on 10 children with VS over a period of 26 years. Compared to more than 2000 skull base surgeries during the same time, sporadic pediatric VS showed a higher rate of preoperative hearing loss as well as other cranial nerve deficits. The predominant sex was male contrary to the adult population [2].

To date, the largest cohort of pediatric sporadic VS reported on 15 children with a mean age of 16.5 years. Mostly, hearing loss, followed by headaches, vertigo, ataxia, and tinnitus were the presenting symptoms. Residual tumors might show a higher rate of regrowth than those in adults [18].

We report on a large cohort of sporadic VS in children and discuss our observations with recent literature.

## 2. Materials and Methods

### 2.1. Inclusion

We included all patients in care at our institution between 2003 and 2021 harboring a sporadic vestibular schwannoma and younger than 21 years of age at the time of diagnosis. A total of 1635 operative cases were identified; 1534 were excluded for being older than 21 years. Of the remaining 101 patients, 73 with NF2 were excluded. Moreover, 28 patients without clinical signs of NF2 (negative family history, no sign of bilateral VS, meningioma, other schwannomas, ependymomas, brainstem ischemia, juvenile cataract, retinal hamartoma, scoliosis) remained for further analysis. Figure 1 shows the inclusion flow chart.

### 2.2. Analysis

A retrospective review of clinical parameters, surgical data, and outcomes was performed. All procedures were performed via retrosigmoid approach either in supine or semi-sitting position by the senior authors (M.T. and M.S.) Preoperative MRI and CT scans were assessed with consideration of cystic formation and bone erosion. Bone erosion was defined as bone destruction surrounding the tumor exceeding the usual internal auditory canal expansion.

### 2.3. Testing for NF2

Clinical work-up of patients with sporadic VS younger than 16 years routinely includes family history, physical examination for peripheral nerve tumors and ophthalmological examination to rule out specific findings such as juvenile cataract. Furthermore, genetic testing is performed of peripheral blood and performed via mutation analysis in the NF2 gene at an external genetic facility specialized in NF-diagnostics.

### 2.4. Tumor Volumetry

Pre- and post-operative MRIs were used to define tumor volumes via volumetry as described before [19]. Initial tumor size as well as—if applicable—residual tumor volumes were examined.

### 2.5. Statistical Analysis

Statistics were analyzed using SPSS Statistics 25 (IBM, New York, NY, USA). Continuous data were presented as mean (±Standard deviation), whereas categorical data were shown as percentages. Continuous variables were tested for equality of variances by Levene’s test. Normal distributed parametric variables with equal variances were compared using the unpaired or paired t-test, otherwise Mann-Whitney U test was performed. Nominal variables were tested with Fisher’s exact test. *p* values < 0.05 were regarded as significant.

This study was performed in line with the principles of the Declaration of Helsinki. Institutional review board approval was granted by the Ethics Committee of University of Tuebingen (705/2018BO2).

## 3. Results

There was a slight predominance of female sex (60.7% *n* = 17). Mean and median age at diagnosis were 16.0 ± 3.3 years (range 6–21 years). Table 1 shows basic patient characteristics.

### 3.1. Pre- and Post-operative Assessment

#### 3.1.1. Symptomatology and Tumor Size

Initial symptoms were predominantly hearing loss and/or tinnitus (67.0% *n* = 19), followed by vertigo/ataxia (17.9% *n* = 5), facial palsy (7.1% *n* = 2) and headache (7.1% *n* = 2). Delay between the appearance of initial symptoms and the diagnosis ranged from 1 to 48 months (mean 10.4 ± 12.5 months).

Tumor size at the time of diagnosis was classified via Hannover grading: T1 (*n* = 8; 28%), T2 (*n* = 7; 25%), T3a (*n* = 5; 18%), T4a (*n* = 3; 11%), T4b (*n* = 5 18%). Tumor volume at the time of diagnosis ranged from 0.085 cm³ to 19.331 cm³ (mean 3.15 ± 4.78 cm^3^).

#### 3.1.2. Growth Rate and Imaging

In 8 cases, more than 1 preoperative MRI scan after the initial imaging was available, allowing us to calculate preoperative tumor growth rates. Tumor volume progression was observed in all 8 cases. The fastest growth rate was 0.22 cm³/month whereas the slowest was 0.009 cm³/month (mean 0.058 ± 0.0072 cm³/month). Relative tumor growth varied from 1 to 18%/month (mean 8.9 ± 5.6%).

Preoperative MR imaging revealed cystic morphology in four cases (14.2%). Notably, all 4 children presented with large tumors T4a and T4b. The patient with the fastest growth rate (0.22 cm³/month, 14%/month), on the other hand, exhibited neither cystic tumor formation nor bone erosion.

Preoperative CT scans were routinely performed to identify emissary veins, position of the jugular bulb, relation of internal auditory canal to the vestibular organ as well as petrous bone air cells. Significant and unusually extensive bone erosion was seen in 3 cases (10.7%). These patients all harbored T4a and T4b tumors. In one of these children, bone erosion and cystic morphology of the tumor presented simultaneously.

Intraoperatively, strong vascularization was seen in only one case. This case presented with cystic tumor formation but without bone erosion. Table 2 summarizes pre- and post-operative growth rates.

#### 3.1.3. Hearing

Preoperative Gardner & Robertson scale (GR) was available for all 28 patients: 1 (28.5%, *n* = 8), 2 (18%, *n* = 5), 3 (39.5%, *n* = 11), 4 (14% *n* = 4). Postoperative GR was available for 21/23 operated patients: 1 (14%, *n* = 3), 2 (10%, *n* = 2), 3 (28.5%, *n* = 6), 4 (38%, *n* = 8), 5 (9.5%, *n* = 2).

Preoperative AEP was performed in 24 patients. Potentials were normal in 17.9% (*n* = 5), affected in 46.4% (*n* = 13) and not measurable in 21.4% (*n* = 6).

Postoperative AEP was available in 20/23 patients; it was reported as normal in 2 patients (7.1%), affected in 6 (21.4%) and not measurable in 12 (42.9%).

Thus, 44.4% (8/18) percent of patients retained auditory function unchanged, 6 patients lost functional hearing. There was a clear correlation of hearing preservation to preoperative quality of hearing.

Tinnitus was present in 11 patients preoperatively (39.3%). At 3-month follow-up, tinnitus was reported in just 3 patients (10.7%). Vertigo was present in 9 patients preoperatively (32.1%) and only in 3 at 3 month follow up (10.7%). Ataxia was present in 3 patients preoperatively (10.7%) and in 2 at the 3-month follow-up (7.1%). Table 2 summarizes pre- and post-operative hearing function.

#### 3.1.4. Facial Nerve

Pre-interventional H&B (House & Brackman) grading of facial nerve function was available for all 28 cases: H&B 1 in 22 cases (87.6%), H&B 2 in 2 (7.1%), H&B 3 in 3 (10.7%) and H&B 4 in 1 patient (3.6%). All patients with T1-T3 tumors presented with H&B 1. On the contrary, almost all patients with large tumors T4a or T4b presented with facial paresis of variable degree H&B 4 (*n* = 1) H&B 3 (*n* = 3), H&B 2 (*n* = 2) and H&B 1 (*n* = 2).

Of the 23 operative cases, intraoperative facial nerve affection was detected via MEP-decrease in 11 (48%) children. However, only 4 (17%) patients (T3a *n* = 3; T4b *n* = 1) showed a postoperative increase in preexisting or new impairment of facial nerve function.

Postoperative facial nerve function was graded as H&B 1 in 14 patients (61%), 2 in 2 (9%), 3 in 4 (17%), and 4 in 3 patients (13%). At 3-month follow-up: H&B 1 in 17 (74%) children, H&B 2 in 1 child (4%), H&B 3 in 3 children (13%) and 4 in 2 children (9%). At 9-month follow-up: H&B 1 in 17 children (74%), H&B 2 in 4 children (17%), H&B 3 in 2 children (9%) and 4 in 0 children. Figure 2 and Table 2 summarize facial nerve function pre- and postoperatively.

#### 3.1.5. Other Cranial Nerves

Trigeminal nerve affection was present in 2 patients preoperatively (7.1%) and was never an initial symptom. In both children, facial hypesthesia persisted postoperatively. Lower cranial nerve affection was seen in 2 cases preoperatively (7.1%). Postoperatively, one child completely recovered. The other child had persisting slight dysphagia.

### 3.2. Non-Operative Cases

A total of 5 additional children were identified via the pediatric neurosurgery database with sporadic VS but without surgical interventions. The reasons included: small tumor size, radiotherapy, or decision for surgery at another institution. Follow-up was not available.

### 3.3. Grade of Surgery and Effects on Tumor Growth

In all 23 surgical cases, resection was performed via retrosigmoid craniotomy. Depending on tumor size, surgery was carried out either in Jannetta position (*n* = 8: T2 *n* = 5; T3a *n* = 2; T4b *n* = 1) or semi-sitting position (*n* = 15: T1 *n* = 2; T2 *n* = 1; T3a *n* = 4; T3b *n* = 1; T4a *n* = 3; T4b *n* = 4).

#### 3.3.1. Gross Total Resection (GTR)

Gross total resection (GTR) was achieved in 18 out of 23 cases (78%). Follow-up MRI showed no tumor regrowth during follow up (mean 28 ± 28.3 months; range 3–94 months).

#### 3.3.2. Subtotal Resection (STR)

All 5 patients receiving STR (22%) presented with large tumors: T3a *n* = 1, T4a *n* = 1 and T4b *n* = 3. Reasons for STR included: tumor adherence to the facial nerve and incipient intraoperative electrophysiological deterioration in all cases. All patients showed regrowth of residual tumor during follow-up. Two children showed significant regrowth at 12-month follow-up MRI.

In one of the cases with bone erosion, tumor residual increased 1.344 cm^3^ in 12 months (to 64% of initial size), growth rate was 0.122 cm^3^/month (5.3%/month). The other case featured a hypervascular, cystic tumor and increased 0.053 cm^3^ (36%) in size in 12 months (0.0045 cm^3^/month; 3.0%/month).

One of these two cases had an especially aggressive course and underwent re-resection twice. Notably, this patient presented initially with significant bone erosion. A subtotal resection was performed at an external institution. Due to significant tumor progression, subtotal re-resection was performed in our center. However, at 12-month follow-up, extrameatal tumor progression was again detected. Reoperation resulted in GTR.

In all other 3 STR cases, significant regrowth was detected at 24-month follow-up. The increase in size in 1 case with bone erosion was 0.312 cm^3^ (115%) at 24 months (0.013 cm^3^/month; 4.8%/month), in the other 2 cases without bone erosion, 0.08 cm^3^ (0.004 cm^3^/month, 1.15%/month) and 0.05 cm^3^ (0.002 cm^3^/month 0.5%/month). Mean tumor regrowth rate in cases with early regrowth (at 12-month follow-up) was 0.059 ± 0.07 cm^3^/month (mean relative regrowth 49.8 ± 19.5%/month) vs. 0.006 ± 0.006 cm^3^/month (mean relative regrowth 51.9 ± 55.1%/month) in cases with delayed residual tumor regrowth (at 24-month follow-up). Statistical analysis showed no significant difference (*p* = 0.287).

In one STR case, data on pre- and post-operative tumor growth rate was available. The absolute postoperative growth rate was lower after STR than before (0.013 cm³/month vs. 0.06 cm³/month). However, relative postoperative growth rate was higher than preoperative growth rate (4.81%/month vs. 1%/month). There was no statistically significant difference between all available STR postoperative tumor growth rates and all available preoperative tumor growth rates (*p* = 0.345 Wilcoxon-Test).

Three STR cases underwent secondary radiotherapy. Follow-up was available in two cases. In both cases, residual tumor growth arrest and thus tumor control was achieved.

In one case, wait-and-scan strategy was implemented. In this case, after regrowth of 0.08 cm^3^ (27.5%; 0.003 cm^3^/month, 1.15%/month) at 24-month follow-up, residual tumor was stable in size for 5 following years. However, between the 5th and the 8th postoperative year, a growth rate of 0.138 cm^3^ (47.5%; 0.003 cm^3^/month; 1.3%/month) was noted. Figure 3 shows pre- and post-operative MRI sets of two exemplary patients with GTR and STR, respectively.

#### 3.3.3. Complications

Cerebrospinal fluid (CSF) fistulas were observed in 2 cases (7.1%) with the need of lumbar drain. After removing the drain, no further CSF collection was observed. These patients hat no extensive bone erosion. No significant postoperative bleeding, infection or other perioperative adverse events were found.

## 4. Discussion

In our case series of 28 children and adolescents with sporadic VS, symptomatology was diverse. Most children presented with hearing loss and tinnitus. Delay from initial symptoms to diagnosis was relatively short (mean 10.4 ± 12.5 months) compared to intervals reported for adults (mean 33–41 month) [20,21].

Tumors tended to be large given the patients’ young age. Compared to an adult cohort of 2336 patients, a slightly larger mean tumor volume at the time of diagnosis was observed (3.15 cm^3^ vs. 2.63 cm^3^) [22]. Compared to our own NF2 cohort of the same age group, the tumor size difference was even more pronounced (1.52 ± 2.49 cm^3^) [19].

All 8 cases with several pre-operative MRI scans showed volumetric growth between 1 and 18%/month, mean 0.058 cm^3^/month. Varughese et al. described a mean tumor growth rate in adults of 0.19 cm^3^/year (0.016 cm^3^/month) [23]. Our data thus provides a 3-fold higher tumor growth rate for pediatric and adolescent patients with sporadic VS.

Unsurprisingly, cystic tumor morphology and bone erosion is mostly seen with larger tumors. Cystic morphology is reported in 4–23% of VS in adults [24] compared to 14.2% in our cohort

GTR was shown to be effective as no regrowth could be found during 28 months of follow-up. Nevertheless, long-term follow-up is needed to rule out late recurrence even years after surgery as well as secondary diagnosis of NF2 with development of a contralateral VS.

Tumor regrowth after STR was seen in all 5 patients, and growth dynamics did not differ significantly between early (12 month) and late (24 month) detection of regrowth. This regrowth rate of 100% seems to be much more common than in adults. In this population, tumor regrowth after STR is reported in about 17% of cases [25].

We observed a rate of 21% (6/28 cases) of preoperative facial nerve impairment in this specific cohort. Compared to adult series this is rather high, indicating the tendency of pediatric tumors to be more aggressive. Previous case series and reports usually provided average tumor size in cm as compared to our volumetric analysis. Mean age of 16 years was comparable to existing data but presenting symptoms in our case series were milder with reports of up to 90% of elevated intracranial pressure with ataxia or 40% of facial palsy [14,18].

Nevertheless, facial nerve outcome was good at 9-month follow-up and 2/6 patients with pre-operative facial nerve impairment improved post-operatively. Only 2 children with pre-operative facial nerve deficits remained at a H&B grade of 3, with no affection in the rest of the cohort (91%).

Thus, the strategy of accepting a residual tumor to maintain facial nerve integrity and opt for radiosurgery seems to be beneficial to obtain a good cosmetic outcome. This is important in adolescents and young adults, since appearance in this age group is of major importance.

Pre- and post-operative hearing status was unsurprisingly associated with tumor size with larger tumors resulting in worse GR. In this cohort of rather aggressive and often large tumors, hearing preservation is not the main surgical goal. Due to growth of residual tumors, long-term uncertainty of radiotherapy in children and the severe neurological symptoms of facial nerve affection we tend to—if necessary—sacrifice hearing in order to achieve GTR. The cases of STR were due to affection of the facial nerve.

A previous meta-analysis suggested a more aggressive appearance of pediatric VS as compared to their adult counterparts [13]. This was not reproduced in our cohort, whilst single cases showed aggressive growth and clinical deficits.

The presented study is limited by a small sample size even though we report on the largest cohort of sporadic VS in children to date. Furthermore, follow-up duration, clinical appearance and individual pre- and post-operative courses as well as further therapies had a large variance. Thus, the character of this study remains purely descriptive. Only larger cohorts with prospective data can provide unequivocal evidence. Generalizations from our data should be approached cautiously, but the reported tendencies are backed by our data. Lastly, clinically late-onset or mosaic NF2 cannot be excluded with 100% certainty even with genetic testing. Therefore, disease onset in young adulthood could be underdiagnosed in our cohort. Nevertheless, no clinical, genetic, or imaging signs of NF2 could be found in our cohort.

Complication rate was low—comparable to the adult population [26,27]. CSF fistula occurred only in 2/4 patients with pre-operative extensive bone erosion and thus is associated with this feature. Consequently, a prophylactic lumbar drain is an option that might be considered in these situations.

Retrosigmoid craniotomy for VS in adolescents and children is safe and feasible and the outcome is, despite rather large and fast-growing tumors with a high rate of pre-operative facial nerve affection comparable with the adult population. Data on radiotherapy for children with VS is scarce. In a series of 148 sporadic and NF2 cases (mean age 13.9 years), only 8.1% received radiotherapy following surgery and another 4.1% radiotherapy alone [28]. We do not believe that, in this age group with a life expectancy of 60+ years, radiotherapy should be the primary treatment in young patients, late effects have ample time to manifest and the favorable results of this study prove that surgery, with its positive results, should be the primary treatment of choice.

## 5. Conclusions

In conclusion, sporadic VS in children are rare and mostly present with hearing impairment and tinnitus. The pre-diagnostic symptomatic interval seems to be shorter than in adults. Tumor size tends to be larger, and tumors seem to have a faster growth rate as compared to adults. The rate of pre-operative facial nerve impairment also seems higher than in adults. Surgical resection is the primary treatment option. The preservation of facial nerve integrity and function might demand STR. GTR, however, should be the goal of surgical intervention if feasible, as STR does not lead to stable disease and radiotherapy will likely follow within 2–3 years. Therefore, children should be treated in high volume centers which have vast experience in VS surgery to minimize the cases of STR and maximize the chances of GTR with functional facial nerve preservation.

## Figures and Tables

**Figure 1 children-09-00490-f001:**
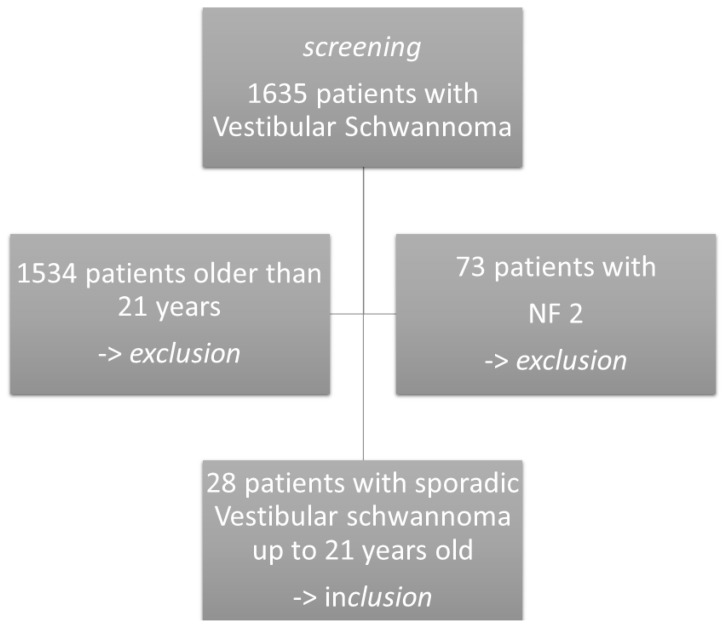
Flow chart of patient inclusion. NF2: Neurofibromatosis Type 2.

**Figure 2 children-09-00490-f002:**
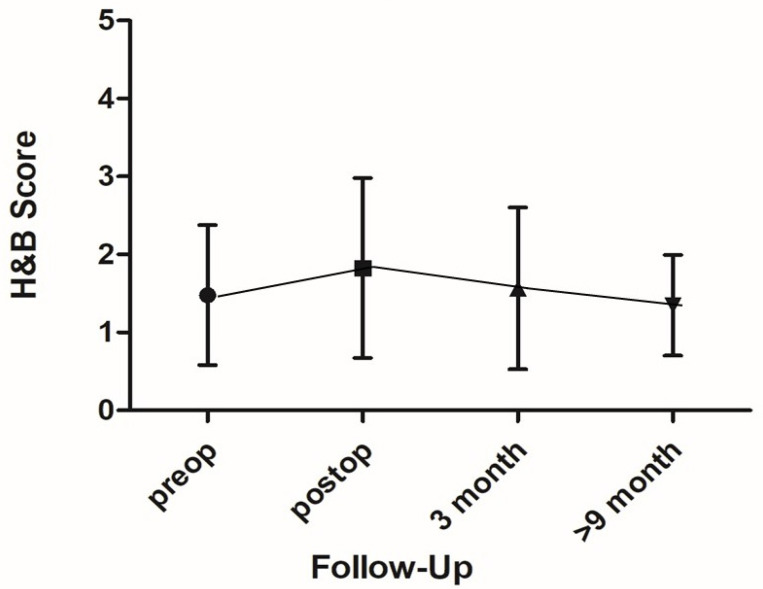
Overview of facial nerve function from preoperatively until 9 months postoperatively, *x*-axis: time, *y*-axis: House-Brackman (H&B)-Score.

**Figure 3 children-09-00490-f003:**
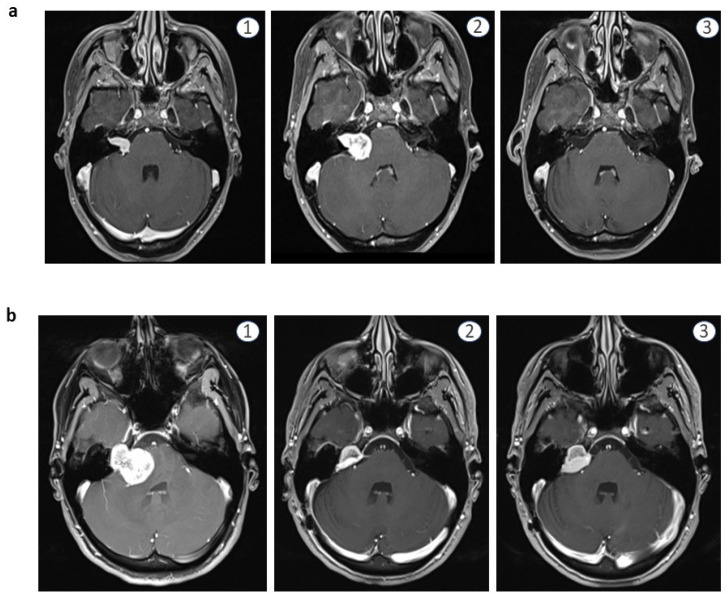
Exemplary MRI images (T1 + contrast) of two patients: (**a**) 17a, male, **1**: first diagnostic MRI 24 months preoperatively, **2**: MRI directly preoperatively, **3**: 12 months follow-up scan without sign of tumor (**b**) 15a, female, **1**: MRI directly preoperatively, **2**: 3 months follow-up scan with residual tumor, **3**: 12-month follow-up scan with progressing tumor.

**Table 1 children-09-00490-t001:** Basic patient characteristics.

#	Sex	Age	Tumor Size (Hannover)	Growth (cm^3^ and %)	Bone Erosion	Cystic Tumor	GR	H&B	GTR/STR	Growth (cm^3^)	Growth in %/Month (postp)	GR	H&B	Recurrence (Months)
			preoperative							postoperative				
1	f	15	T2		-	-	2	1	GTR			3	1	
2	f	17	T4a		-	-	4	1	GTR			4	1	
3	m	16	T2		-	-	3	1	GTR			3	1	
4	f	18	T2	0.207 (54.9%)	-	-	3	0	GTR			3	1	
5	m	13	T4b	1.02 (17%)	yes	-	4	3	STR	0.312	0.013 (4.81%)	4	3	24
6	f	15	T4b		yes	-	3	3	STR	1.344	0.112 (5.35%)	3	3	12
7	m	6	T4a		-	yes	4	4	GTR			4	4	
8	f	16	T4a		yes	yes	4	3	GTR			4	3	
9	f	17	T4a		-	-	1	2	STR	0.08	0.003 (1.15%)	1	2	24
10	m	16	T2		-	-	2	1	GTR			2	1	
11	f	11	T3a		-	-	1	1	STR	0.005	0.002 (0.55%)		4	24
12	m	20	t3a		-	-	1	1	GTR			1	1	
13	m	18	T2		-	-	3	1	GTR			4	1	
14	f	16	T1		-	-	1	1	GTR			2	1	
15	f	15	T4a		-	yes	3	2	STR	0.053	0.004 (3.0%)	4	3	12
16	f	14	T2		-	-	3	1	no surgery					
17	f	13	T1		-	-	2	1	no surgery					
18	m	15	T1	0.672 (364.8%)	-	-	1	1	no surgery					
19	f	15	T3b	1.82 (1260%)	-	-	1	1	no no surgery					
20	f	19	T3a	2.6 (325%)	-	-	3	1	GTR				4	
21	m	20	T3a	1.1 (70%)	-	-	1	1	GTR			5	2	
22	m	19	T3a		-	-	3	1	GTR			4	1	
23	f	12	T1	0.189 (84%)	-	-	3	1	GTR			3	1	
24	f	16	T1	0.204 (96%)	-	-	1	1	GTR			1	1	
25	m	19	T3a		-	-	3	1	GTR			4	1	
26	m	15	T2		-	-	2	1	no surgery					
27	f	21	T2		-	-	3	1	GTR			3	1	
28	f	21	T2		-	yes	2	1	GTR			5	1	

GR: Gardner&Robertson scale, H&B: House&Brackman grading; GTR: gross total resectiom, STR: subtotal resection.

**Table 2 children-09-00490-t002:** Comparison of pre- and post-operative hearing, facial nerve function and tumor growth rates.

	Preoperative			Postoperative		
#	GR	H&B	Growth Rate cm^3^/Month and %/Month	GR	H&B	Growth Rate cm^3^/Month and %/Month
1	2	1	na	3	1	-
2	4	1	na	4	1	-
3	3	1	na	3	1	-
4	3	1	0.023 (6.1%)	3	1	-
5	4	3	0.06 (1%)	4	3	0.013 (4.81%)
6	3	3	na	3	3	0.112 (5.35%)
7	4	4	na	4	4	-
8	4	3	na	4	3	-
9	1	2	na	1	2	0.003 (1.15%)
10	2	1	na	2	1	-
11	1	1	na	na	4	0.002 (0.55%)
12	1	1	na	1	1	-
13	3	1	na	4	1	-
14	1	1	na	2	1	-
15	3	2	na	4	3	0.004 (3.0%)
20	3	1	0.1 (12.5%)	na	4	-
21	1	1	0.22 (14.0%)	5	2	-
22	3	1	na	4	1	-
23	3	1	0.009 (4.0%)	3	1	-
24	1	1	0.017 (8.0%)	1	1	-
25	3	1	na	4	1	-
27	3	1	na	3	1	-
28	2	1	na	5	1	-

Note: #: patient number, na: not available.

## Data Availability

Data is available from the corresponding author.

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
