# Peer review of "Management of Sporadic Vestibular Schwannomas in Children—Volumetric Analysis and Clinical Outcome Assessment"

_children, 2022, doi:10.3390/children9040490_

Round 1

Reviewer 1 Report

I had the opportunity to review the manuscript entitled "Management of sporadic vestibular schwannomas in children – volumetric analysis and clinical outcome assessment". This is a retrospective study of 28 sporadic vestibular schwannomas. The idea is to describe the clinical characteristics, management and outcome of this rare subgroup of patients. This paper provides the community with insights related to the sporadic vestibular schwannomas and its peculiar features. Although this could be a potential good study, there are many major issues.You need a significant writing and editing work by a native English-speaking colleague. 

2. Materials and Methods: please include how do you exclude NF2 in these patients. Did you performed germline anaylisis in all patients to rule out NF2. Please specify the germline methods as NF2 is a difficult gene to analyse and could be misdiagnosed. It is well known that the NF2 diagnosis could be underdiagnosed in children as this disease onset in young adulthood. 

Figure 1: 28 patients with sporadic NF???Do you mean VS

Table 1: please explain meaning of all abbreviations.

Discussion: please compare your finding with previous publications. Maybe you can add a table showing your results with previous case reports/cases series findings. 

Please be realistic with the study limitations as this is a retrospective review with 28 patients with different surgical status and post surgery treatment.

Figure 3: Please add the MRI sequences that you are including in this figure

Is there a role for Bevacizumab as it is in bilateral schwanommas?

Reviewer 2 Report

Well written manuscript about AN in the pediatric population.  Authors noted that tumors tend to be more aggressive than in adults.  Outcomes were sound.  This does add some new insights.

Author Response

na

Reviewer 3 Report

This manuscript presents interesting data on the outcome of sporadic vestibular schwannoma (VS) in children. Since pediatric vestibular schwannoma except for neurofibromatosis type 2 is extremely rare, this study with the largest cohort so far provides an important contribution to the management of the disease. In particular, pre-and post-operative volumetry was well described, and the conclusion that surgical resection is the primary therapy is quite convincing.  

The authors stated that no tumor regrowth was confirmed after gross total resection, although the mean follow-up period was only 28 months. The possibility of recurrence several years after resection would not be negligible, because children naturally have longer survival than adults. Moreover, the cases in that children are diagnosed with neurofibromatosis type 2 after middle to long-term observation exist. The authors should clarify that long-term surveillance is necessary even if GTR is achieved.

Author Response

The discussion was edited to include the statement about long-term surveillance.

Round 2

Reviewer 1 Report

Although this version is much better, i would make some comments:

Materials and Methods: 

  • Please add some comments about the genetic work up you have done in yours patients.  
  • English spell check is requiered.

Author Response

English language editing has been performed and we included a subsection on ruling out NF2 including a comment on genetic testing.